# *Solanum nigrum* Extract and Solasonine Affected Hemolymph Metabolites and Ultrastructure of the Fat Body and the Midgut in *Galleria mellonella*

**DOI:** 10.3390/toxins13090617

**Published:** 2021-09-01

**Authors:** Marta Spochacz, Szymon Chowański, Monika Szymczak-Cendlak, Paweł Marciniak, Filomena Lelario, Rosanna Salvia, Marisa Nardiello, Carmen Scieuzo, Laura Scrano, Sabino A. Bufo, Zbigniew Adamski, Patrizia Falabella

**Affiliations:** 1Department of Animal Physiology and Developmental Biology, Faculty of Biology, Adam Mickiewicz University in Poznań, 61-614 Poznań, Poland; szyymon@amu.edu.pl (S.C.); monikasz@amu.edu.pl (M.S.-C.); pmarcin@amu.edu.pl (P.M.); zbigniew.adamski@amu.edu.pl (Z.A.); 2Laboratory of Electron and Confocal Microscopy, Faculty of Biology, Adam Mickiewicz University in Poznań, 61-614 Poznań, Poland; 3Department of Sciences, University of Basilicata, 85100 Potenza, Italy; filomenalelario@hotmail.com (F.L.); r.salvia@unibas.it (R.S.); nardiellomarisa@gmail.com (M.N.); carmen.scieuzo@unibas.it (C.S.); laura.scrano@unibas.it (L.S.); sabino.bufo@unibas.it (S.A.B.); patrizia.falabella@unibas.it (P.F.); 4Department of European Culture, University of Basilicata, 75100 Matera, Italy; 5Department of Geography, Environmental Management & Energy Studies, University of Johannesburg, Johannesburg 2092, South Africa

**Keywords:** *Galleria mellonella*, glycoalkaloids, *Solanum nigrum* extract, solasonine, hemolymph, polyols, proline, ultrastructure, fat body, botanical insecticides

## Abstract

Glycoalkaloids, secondary metabolites abundant in plants belonging to the Solanaceae family, may affect the physiology of insect pests. This paper presents original results dealing with the influence of a crude extract obtained from *Solanum nigrum* unripe berries and its main constituent, solasonine, on the physiology of *Galleria mellonella* (Lepidoptera) that can be used as an alternative bioinsecticide. *G. mellonella* IV instar larvae were treated with *S. nigrum* extract and solasonine at different concentrations. The effects of extract and solasonine were evaluated analyzing changes in carbohydrate and amino acid composition in hemolymph by RP-HPLC and in the ultrastructure of the fat body cells by TEM. Both extract and solasonine changed the level of hemolymph metabolites and the ultrastructure of the fat body and the midgut cells. In particular, the extract increased the erythritol level in the hemolymph compared to control, enlarged the intracellular space in fat body cells, and decreased cytoplasm and lipid droplets electron density. The solasonine, tested with three concentrations, caused the decrease of cytoplasm electron density in both fat body and midgut cells. Obtained results highlighted the disturbance of the midgut and the fat body due to glycoalkaloids and the potential role of hemolymph ingredients in its detoxification. These findings suggest a possible application of glycoalkaloids as a natural insecticide in the pest control of *G. mellonella* larvae.

## 1. Introduction

Plant derivatives have been extensively studied for their efficacy against many insect species to prove their insecticidal properties [1,2,3,4]. Black nightshade (*Solanum nigrum*) is a member of the Solanaceae family, the same as potato (*Solanum tuberosum*) or tomato (*Solanum lycopersicum*). These plants contain secondary metabolites, glycoalkaloids, that may prevent the attacks of herbivores. *S. nigrum* contains the highest amounts of two main glycoalkaloids: solasonine and solamargine. Both *S. nigrum* unripe berries extract and pure glycoalkaloids were tested previously for insecticidal activity on coleopteran insects, confirming their sublethal effect [2,5,6]. However, products from *Solanaceae* plants tested on lepidopteran models showed lethal, sublethal, or no effects [7]. *S. tuberosum* extract and α-solanine caused developmental changes, decreased survivorship, fecundity, and fertility, and increased oxidative stress [8,9]. No studies are available about the effects of *S. nigrum* extract and its components on the greater wax moth, *Galleria mellonella* Linnaeus (Lepidoptera), a pest of the beehives. We investigated its properties on this pest, which is a common model in biomedical studies [10]. When the bee population is weakened, and the maintenance conditions are poor, *G. mellonella* may cause a severe invasion. It damages the brood comb, limits the bee population, and feeds on bee products, causing crop losses [11]. The insects also feed on the wax that remains on the bottom of a hive, which can be treated with biological insecitcides of low toxicity to bees and short persistence. Therefore, the control of this species is difficult. The majority of synthetic insecticides cannot be used by breeders, as they are toxic to bees and cause pollution of bee products. As alternatives to synthetic pesticides, natural substances can be less harmful to bees and bee product consumers. In this study, we tested the influence of dietary *S. nigrum* extract and solasonine in three different concentrations on the physiology of *G. mellonella*. The aim of this study was to determine whether the extract and its main glycoalkaloid, solasonine, can cause lethal or sublethal effects on *G. mellonella*. This step is crucial for further studies on the selectivity of tested substance to the pest and the bees. If the positive effects can be observed, the studies on the proper strategies of application may be developed. Some of the checked parameters were sugars, polyols, and proline levels in hemolymph, the insect circulating fluid. The hemolymph is a sensitive marker of changes in the insect organism; the content of particular components in this tissue is under control of the fat body and may depend on various conditions such as temperature, developmental stage, and metabolic state [12,13,14]. Still, not much is available about maintaining homeostasis of sugars, polyols, and proline during intoxication and, hence, other possible functions of these compounds. As the second parameter, we studied the ultrastructure of the midgut epithelium and the fat body cells responsible for digestion and detoxification, respectively. These two tissues are exposed to both synthetic and natural pesticides entering the bodies of insects with food. Previous research proved that they undergo ultrastructural alterations when exposed to substances of the insecticidal activity. The malformations of biological membranes, nuclei, and mitochondria have been observed, sometimes as a response to low concentration of these substances [1,2,8]. Therefore, the ultrastructural alterations can be used as the biomarkers of sublethal toxicity. Hemolymph is a link between these two tissues. It contains metabolites of which levels change due to many physiological conditions and can be markers of ongoing processes. 

## 2. Results

### 2.1. The Effects of S. nigrum Extract and Pure Glycoalkaloids on Sugars, Polyols, and Proline Concentrations in the Hemolymph

#### 2.1.1. Chromatography

During the HPLC analysis performed on hemolymph samples derived from control larvae and larvae fed on the extract and on the pure glycoalkaloids at different concentrations, 19 peaks appeared representing 15 compounds, among which 7 were identified (Figure 1). According to their appearance on the chromatogram, they were erythritol, proline, sorbitol, glucose, mannitol, inositol, and trehalose.

#### 2.1.2. Polyols

In the hemolymph of *G. mellonella*, it was possible to identify four polyols: erythritol, mannitol, inositol, and sorbitol. The level of total polyols in the hemolymph increased after application of all tested substances compared to the control (mean ± SEM = 0.5 ± 0.11 µg/µL) (Figure 2). The most significant change was due to SolA (1.0 ± 0.19 µg/µL), but none of these changes were statistically significant. Different concentrations of each polyol were observed after HPLC analysis. For example, the lowest and the highest concentrations of erythritol in the hemolymph were 0.01 µg/µL and 0.15 µg/µL, respectively, while inositol’s lowest concentration was 0.11 µg/µL, and the highest was 0.46 µg/µL. 

Moreover, separated polyols differed in appearances, such as mannitol after treatment with the extract (Ext), and solasonine in the same concentration as in the extract (SolB) was not found in tested samples (Figure 2). The mean value for the control was equal to 0.01 ± 0.003 µg/µL. An increased concentration was observed after the application of SolC (0.5 ± 0.17 µg/µL). Sorbitol concentration in the hemolymph of control larvae was characterized by a wide dispersion, which decreased after applying solasonine in the lowest concentration (SolC). The level of erythritol increased after the treatment with all substances compared to the control (0.04 ± 0.008 µg/µL, *n* = 9), but only after the application of the Ext (0.12 ± 0.015 µg/µL) were changes statistically significant (*p* = 0.0069, *n* = 5). The level of inositol was similar to the control level (0.2 ± 0.02 µg/µL) in the hemolymph of larvae tested with all used substances except SolA (0.4 ± 0.03 µg/µL).

#### 2.1.3. Sugars

Two sugars, glucose and trehalose, were identified in the hemolymph samples of *G. mellonella* larvae (Figure 3). The total level of free sugars remained similar to the control larvae (11.3 ± 0.82 µg/µL) with a slight decrease after an application of the Ext (7.8 ± 0.79 µg/µL). Glucose is a reducing sugar; hence, it is present in the hemolymph in small quantities. The mean concentration of this sugar in the hemolymph was 2.1 ± 0.36 µg/µL in the control larvae. After feeding larvae with a diet containing Ext and SolC, the glucose levels in the hemolymph decreased (0.98 ± 0.36 µg/µL and 1.1 ± 0.23 µg/µL, respectively) compared to the control and increased after the treatment with SolB (4.0 ± 0.59 µg/µL). The changes were significant between the glucose levels after application of Ext and SolC compared to SolB (*p* = 0.0049, *n* = 5); however, no differences were detected for the control larvae. Trehalose was found in large quantities (up to 13.25 µg/µL in the control larvae). The mean value for the control larvae was 9.2 ± 0.89 µg/µL and remained similar after the application of all tested substances, with the lowest level after application of the Ext (6.9 ± 0.88 µg/µL).

#### 2.1.4. Proline

One amino acid, proline, was identified in the hemolymph of *G. mellonella* larvae. Its maximum concentration in the hemolymph reached 4.69 µg/µL in the control larvae (Figure 4), where the mean value was equal to 3.6 ± 0.28 µg/µL. The application of all tested substances caused the decrease of proline concentration in the hemolymph, but only the application of Ext and SolB caused significant changes (2.11 ± 0.3 µg/µL and 2.11 ± 0.18 µg/µL, respectively; *p* = 0.0132, *n* = 5). 

Results containing the effects on sugars, polyols, and proline concentrations in the hemolymph are summed up in Table 1 (below). 

### 2.2. The Effects on the Ultrastructure 

#### 2.2.1. The Fat Body

The ultrastructure of the fat body cells from *G. mellonella* larvae is shown on the exemplary electronograms (Figure 5). In the control cells, the nucleus is oval-shaped and contains evenly distributed patches of heterochromatin. The cytoplasm is rich in big and homogenous lipid droplets frequently filled with stored, more electron-dense lipids formed in a centered patch. In the cytoplasm, numerous electron-dense mitochondria are present, and glycogen granules (Figure 5, No 2) are scattered. An endoplasmic reticulum is present in some areas of the cell (Figure 5, No 3). The cell membranes adhere tightly (Figure 5, No 2).

*S. nigrum* extract added to the nourishment caused the appearance of intracellular space between cells (Figure 5, Nos 4 and 5) and altered electron density of the cytoplasm (Figure 5, Nos 4 and 6); however, some cells showed increased density of cytoplasm (Figure 5, No 6). Additionally, in some of the lipid droplets, the centered patch of fat shrunk (Figure 5, No 5). Pure solasonine (SolA) caused the decrease of cytoplasm electron density around areas with glycogen granules (Figure 5, No 7). Similarly, as the Ext, SolB caused the decrease of lipid droplets homogeneity (Figure 5, No 8). The lowest used solasonine concentration (SolC) caused the decrease of electron density of cytoplasm (Figure 5, No 9) and nuclei (Figure 5, No 9).

#### 2.2.2. The Midgut

The ultrastructure of *G. mellonella* midgut cells (control) revealed three types of cells. The first type was digestive cells of epithelium (Figure 6, No 1) with round, elongated nucleus and microvilli and electron-dense cytoplasm abundant in mitochondria. The second type of cells protruded to the gut lumen (Figure 6, No 2) with electron-lucent cytoplasm rich in the rough endoplasmic reticulum. The third type of cell was the goblet cell with characteristic mitochondria in the microvilli (Figure 6, No 3) observed in Lepidoptera [15]. The application of Ext caused the appearance of electron-lucent areas close to microvilli, probably signs of vacuolization (Figure 6, No 4) with loss of contact between cells (Figure 6, No 4). In the bulging cell to the gut lumen described above, an increase in the amount of endoplasmic reticulum was observed (Figure 6, No 5) with vacuolization on the apical part (Figure 6, No 5). Additionally, other digestive cells possessed areas of electron-lucent cytoplasm (Figure 6, Nos 6 and 7). In the midgut cells treated with a high concentration of solasonine (SolA), different changes in the ultrastructure such as vacuolization in the vicinity of the endoplasmic reticulum (Figure 6, Nos. 8 and 9) were observed. The cytoplasm density decreased significantly (Figure 7, Nos 1–3) when the solasonine (SolB) was applied in the nourishment in the same concentration as in the extract. In some cells, a strong vacuolization could be observed (Figure 7, No 2). Doubled decreased concentration of solasonine (SolC) caused an increase in the electron density of the cell cytoplasm (Figure 7, Nos 5 and 6) with the areas of vacuolization (Figure 7, Nos 5 and 6), however, in a part of the cells, the cytoplasm was electron-lucent (Figure 7, No 4).

## 3. Discussion

The chemical composition of *G. mellonella* hemolymph has been studied before and shows differences depending on the method used for the measurements [16,17]. However, the function of particular ingredients and their appearance under various conditions are not well understood. In our studies, total polyols concentration in the hemolymph increased slightly after applying tested substances (Figure 2). Previously obtained data suggest that polyols in insect hemolymph may play a protective role during stress conditions such as cold [18] or heat [19]. When specific polyols were divided into separate compounds, the increase and the decrease of levels of particular compounds were observed, suggesting their different functions. Erythritol, a four-carbon sugar alcohol, was found in all tested larvae. The mean values of its level in the hemolymph of larvae treated with the *S. nigrum* extract (Ext) and three solasonine concentrations were higher than the mean value of the control. The highest increase was observed in the level of erythritol when the Ext was applied (Figure 2). The function of this polyol in *G. mellonella* is unknown; however, it was reported as a signal particle in defense mechanisms in *Drosophila suzukii* [20] and a factor regulating osmotic pressure [21,22]. Erythritol also has membrane protecting properties during oxidative stress [23], suggesting a potential membrane disturbing glycoalkaloid action reported in the literature [24]. Other polyols, such as sorbitol and mannitol found in *G. mellonella* larvae, may function as enzyme and cell membrane stabilizers [25]. It was shown that glycoalkaloids such as α-solanine as well as plant extracts obtained from *Solanaceae* family members added into the diet may lead to induction of oxidative stress in *G. mellonella* [8,9]. Hence, we hypothesize that erythritol, of which levels increased significantly in the hemolymph of larvae treated with the Ext, plays a protective role against free radicals produced during intoxication. Trehalose is the main disaccharide found in the insect hemolymph as well as in *G. mellonella* [16,26]. Its concentration in the hemolymph can vary under such conditions as developmental stage, type and amount of ingested food, and other physiological conditions [27,28]. The increased level of trehalose in the hemolymph can be observed in insects under temperature stress. Similarly, as in polyols, trehalose is considered a cell membrane protector during osmotic stress and a preservative [29]. However, its level in larvae used in the studies slightly decreased compared to the control, suggesting other crucial roles of trehalose, such as energy storage used for detoxification. Glucose levels in the hemolymph of larvae treated with Ext and solasonine (SolA, SolB, and SolC) did not show significant changes compared to the control (Figure 3), while significant differences were noted between SolB and Ext and between SolB and SolC. The concentration of glucose in the hemolymph can change due to the feeding state of an insect or a type of diet. In this case, glucose is converted to other sugars and polyols, e.g., sorbitol in the polyol pathway [30]. We do not exclude the possibility of disrupting enzymes of these pathways by alkaloids; however, more studies should be conducted. 

Proline was the most abundant amino acid in the hemolymph of *G. mellonella* larvae according to Killiny et al. [16]. This amino acid is an energy resource associated with muscles as a compound with high solubility and energy yield [31]. In our studies, its level decreased after feeding larvae with Ext and SolA, SolB, and SolC, suggesting its additional function as a source of energy in detoxification. Similar changes were observed when Ext and SolB were applied, which shows the potential main action of solasonine in the Ext. 

The observed changes in the ultrastructure of the fat body and the midgut suggest the disturbance of these tissues by glycoalkaloids. The most frequent changes observed in the fat body cells were the decrease in electron density of cytoplasm, the increase of intracellular space, and the lysis of lipid droplets content. Vacuolization of cytoplasm in the fat body cells was observed in our previous studies on the *S. nigrum* extract and pure solasonine on *Tenebrio molitor* [2] and in the studies on the tomato cherry and the potato leaf extracts on another Lepidoptera, *Spodoptera exigua* [32]. The literature data confirmed changes in lipid droplets observed after glycoalkaloids application, pointing to disturbances by lipid peroxidation and interactions with lipids [33]. Other data show that glycoalkaloids affect the fat body cells by the increase of production of antioxidative enzymes [8] and reactive oxygen species (ROS) [34], which can be the leading cause of the observed ultrastructural changes of cells; however, further analysis should be conducted. In the midgut, the changes were observed in the electron density of cytoplasm with areas of vacuolization showing similar potential causes, such as disturbances in the oxidative balance. These changes were more significant in the proximal area of the rough endoplasmic reticulum, suggesting stress within these structures. Vacuolization in the cytoplasm of the midgut cells corresponds with our previous results on the *S. nigrum* extract on *T. molitor* [2]. The observed malformations suggest a disturbance of biological membranes and an imbalance of osmotic conditions within the cytoplasm. Indeed, glycoalkaloids bind to sterols in biological membranes. They may form complexes that affect the permeability of membranes and disturb homeostasis within cells, including osmotic conditions, when cytoplasm content leaks out or the physiological fluids leak into cells [35,36]. The glycoalkaloids may act synergistically, explaining more significant malformations observed within tissues exposed to extracts than to the single alkaloid. For example, solasonine synergistically increases the disruption of liposomes caused by solamargine [24]. Considering the quality and the concentrations of applied substances both in the fat body and in the midgut, the Ext caused the most noticeable changes. A decrease of proline level and an increase of erythritol level in the hemolymph were detected. This research confirms the previous findings about synergistic action of other extract ingredients [2,6]. A very interesting aspect would be the evaluation of the effects of Solanaceae extracts used as potential bioinsecticides on bees, as few studies were carried out. Kasiotis and colleagues tested alkaloids of the Nicotiana glauca Graham (Solanaceae) on bees, finding low levels of toxicity causing 16–18% of mortality [37]. Similar effects were detected when bees took high doses of nicotine, a typical alkaloid of the Solanaceae plant family, including tomato, potato, green pepper, and tobacco. Indeed, no remarkable effects were detected in adult or larval survival or egg hatching [38]. Further studies need to confirm that *S. nigrum* Ext and/or solasonine could have minimal impact on bees, enhancing the possibility to use them as bioinsecticides.

## 4. Conclusions

*S. nigrum* extract and pure solasonine treatments showed sublethal effects on *G. mellonella* larva. The results showed that glycoalkaloids present in the extract and pure solasonine may interact with the midgut and the fat body cells, generating damages. Changes in the hemolymph metabolites profile suggest disturbances in the larvae metabolism. The above-mentioned effects prove the toxic mode of action of glycoalkaloids on the insect organism and may lead to their weakening and limiting of the population. Conducted experiments show the possibility of potential usage of glycoalkaloids as bioinsecticides and indicate other unknown functions of hemolymph metabolites during intoxication and their mechanisms of action.

## 5. Materials and Methods

### 5.1. Larvae

*G. mellonella* fourth instar larvae were selected from the breeding available at the laboratory of Insect Physiology and Molecular Biology, University of Basilicata (Italy). Larvae were reared on an artificial diet [39] under controlled conditions: temperature 29.0 ± 1.0 °C, relative humidity of 70 ± 5% in complete darkness. 

### 5.2. Nourishment Preparation

According to recipe [39], the nourishment was prepared as follows: in a sterilized beaker, 500 g of white wheat flour, 250 g of whole wheat flour, 250 g of whole meal corn flour, and 250 g of milk powder were combined and autoclaved. Then, 225 g of beeswax and 500 g of organic honey were heated at 100 °C until boiling and allowed to cool at room temperature. Then, 125 g of Brewer’s yeast and 250 g of pure glycerin were mixed thoroughly with the liquids and the sterile powders until the mixture crusted. Once prepared, the culture medium was stored at 4 °C to avoid dry conditions.

### 5.3. Extract and Solasonine Preparation

The extract from *S. nigrum* berries was prepared at the laboratory of professor Sabino A. Bufo, Potenza, Italy. The voucher specimen was deposited at Herbarium Lucanum (HLUC, Potenza, Italy) with ID Code 2320. The extract was prepared with the method previously described [2,40]. In the experiment, one extract concentration was used, 0.5 mg per 1 g of the diet. Solasonine was purchased from a commercial supplier (Sigma Aldrich) and was diluted in a 1% formic acid solution. Solasonine concentration was 149.3 ± 11.3 mg/100 g of the dry extract. Three solasonine solutions were prepared according to their concentration in the extract:SolA: two times higher than the extract concentration (1.49 ×10^−3^ mg/1 g of the diet);SolB: equal to the extract concentration (7.465 × 10^−4^ mg/1 g of the diet);SolC: two times lower than the extract concentration (3.825 × 10^−4^ mg/1 g of the diet).

The solasonine concentrations used were the same as the compound determined in the extract, allowing us to compare whether obtained results were the effect of solasonine or the other extract ingredients or the effect of the interaction between solasonine and other compounds.

### 5.4. Experiment

In each plastic tube containing 10 g of the diet mixed with tested substances, six fourth instar larvae were placed. Control larvae were placed in a plastic tube containing 10 g of diet mixed exclusively with a solvent (1% formic acid). The tubes with insects and the prepared diet were kept in the same conditions as in the breeding. After three days of treatment, larvae were taken out, and the samples of the midgut, the fat body, and the hemolymph were collected. The experiment was conducted in 3 repetitions for each concentration in 5 to 10 biological replicates in the case of hemolymph studies and in 3 biological replicates in the case of transmission electron microscopy.

### 5.5. Samples Collection for RP-HPLC

Larvae fed with the extract, larvae fed with the pure glycoalkaloids at different concentrations, and control larvae were chosen randomly and anaesthetized with carbon dioxide. The leg from the third pair was cut off, and 2 μL of the hemolymph (10 μL per sample from 5 larvae) were collected with an automatic pipette into 500 μL of 70% ethanol. Samples were shaken and centrifuged for 10 min at 10,000× *g*, then filtered with Millex Samplicity^TM^ filters with pore size 0.20 μm (Merck Millipore, Germany) into the glass bottles for HPLC. For each concentration, 5–6 samples were collected, and 10 samples were collected for the control.

### 5.6. Qualitative and Quantitative Determination of Hemolymph Components

The changes in carbohydrates and amino acids composition in the hemolymph were analyzed by the reverse-phase high-performance liquid chromatography (RP-HPLC) according to methods previously described by Słocińska et al. [14]. For separation, a Dionex Ultimate 3000 (USA) chromatographic system comprising a dual-pump programmable solvent module and a Corona Charged Aerosol Detector (CAD) was used. The samples were analyzed on an Asahipak NH2P-50 4E column (250 × 4.6 mm, Shodex, Japan) and eluted with multi-step gradient of ACN concentration and flow rate as follows: 0–5 min (86%, 1 mL/min), 5–10 min (83%, 1 mL/min), 10–20 min (81%, 1 mL/min), 20–38 min (81%, 1.4 mL/min), and 38–40 min (86%, 1 mL/min) at 38 °C. The standards of analyzed substances (Merck, Germany) as solutions in 70% ethanol at a concentration of 1 mg/mL were used to obtain a standard curve.

### 5.7. Samples Collection for TEM

Fat body and midgut portions were isolated and cleaned from undesirable tissue debris and placed in a glutaraldehyde solution (4%) on cacodylate buffer (0.175 M) for 2 h, postfixed with osmium tetroxide (2%) for 2 h, dehydrated with increasing concentrations of ethanol solutions, and finally embedded in Spurr resin (Electron Microscopy Sciences, Hatfield, PA, USA). Ultrathin sections were cut with a Leica ultramicrotome and stained with uranyl acetate and lead citrate. Samples were observed under the transmission electron microscope JEOL 1200EX II JEM (JEOL, Tokyo, Japan).

### 5.8. Statistical Analysis

The analysis was conducted with a software GraphPad Prism 5.01 (GraphPad Software, Inc., San Diego, CA, USA). In the studies on the hemolymph composition, we used the Shapiro–Wilk normality test to check the normality distribution. Next, one way ANOVA with Dunn’s multiple comparison test was used for statistical significance of the differences. The values were presented with ±SEM.

## Figures and Tables

**Figure 1 toxins-13-00617-f001:**
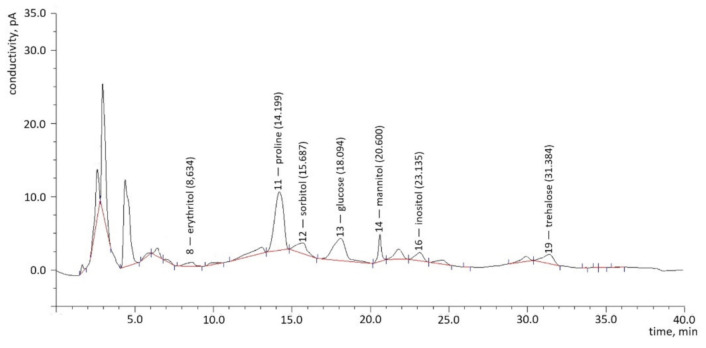
An exemplary chromatogram presenting identified compounds and their retention time from one of the hemolymph samples.

**Figure 2 toxins-13-00617-f002:**
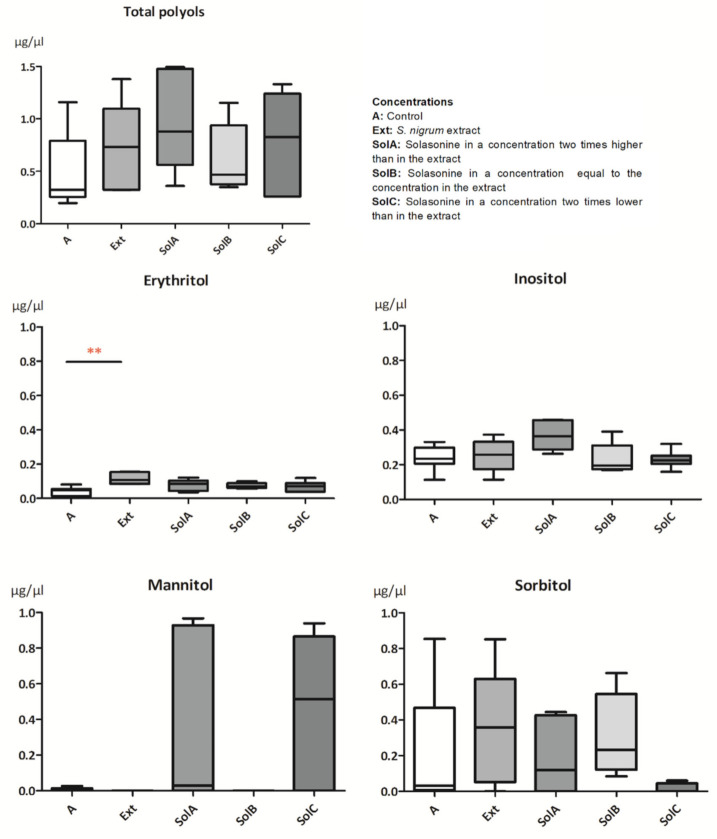
Concentrations of polyols in the hemolymph of *G. mellonella* larvae treated with *S. nigrum* extract and three concentrations of solasonine corresponding with its concentration in the extract. Data are presented as mean ± SEM and compared by Dunn’s multiple comparison test. Statistically significant differences between samples are indicated with asterisks, ** *p* ≤ 0.01, *n* ≥ 5.

**Figure 3 toxins-13-00617-f003:**
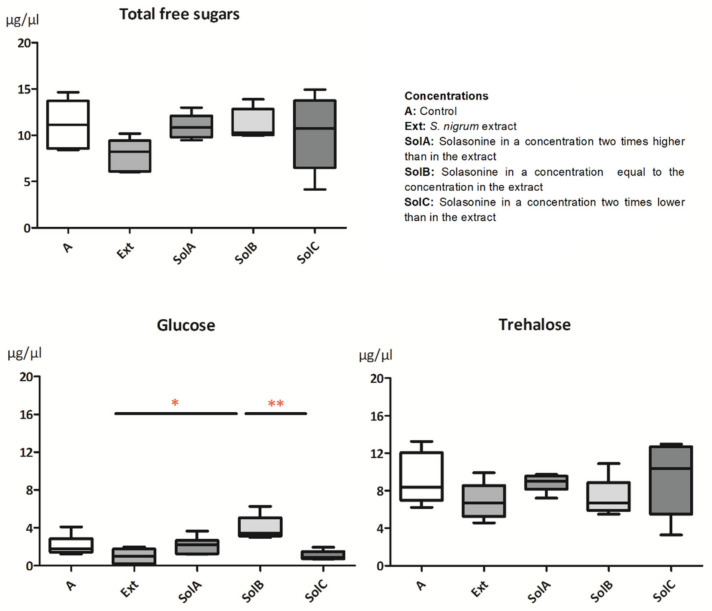
Concentrations of glucose and trehalose in the hemolymph of *G. mellonella* larvae treated with *S. nigrum* extract and its main glycoalkaloid, solasonine, in three different concentrations corresponding with its concentration in the extract. Data are presented as mean ± SEM and compared by Dunn’s multiple comparison test. Statistically significant differences between samples are indicated with asterisks, * *p* ≤ 0.05, ** *p* ≤ 0.01, *n* ≥ 5.

**Figure 4 toxins-13-00617-f004:**
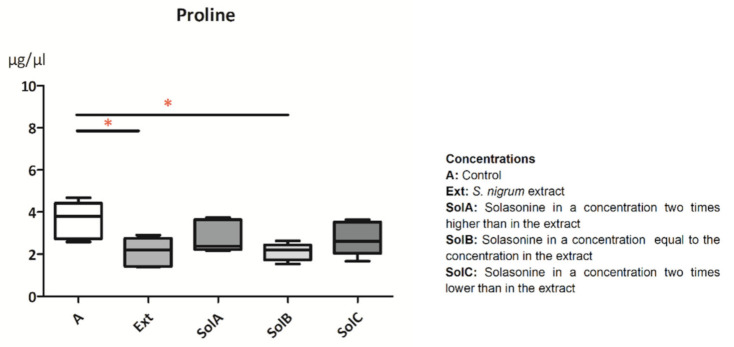
Proline levels in the hemolymph of *G. mellonella* larvae treated with the *S. nigrum* extract and its main glycoalkaloid, solasonine, in three different concentrations. Data are presented as mean ± SEM and compared by Dunn’s multiple comparison test. Statistically significant differences between samples are indicated with asterisk, * *p* ≤ 0.05, *n* ≥ 5.

**Figure 5 toxins-13-00617-f005:**
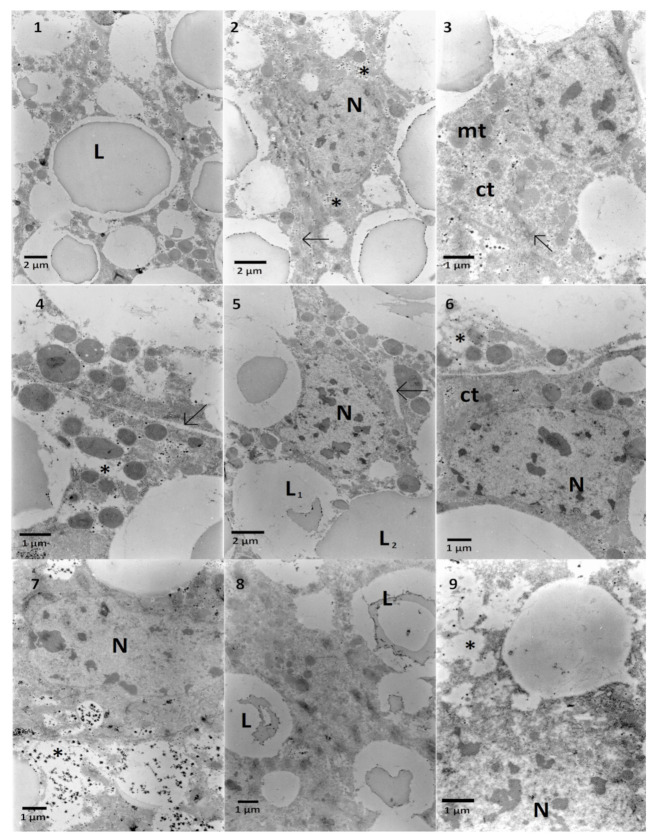
Fat body cells of *G. mellonella* larvae. 1–3—Control cells. Homogenous lipid droplets (**L**) can be observed. Each cell contains a nucleus (**N**) and cytoplasm (ct) with mitochondria (mt), glycogen granules (asterisk), and endoplasmic reticulum (arrow). 4–6—*S. nigrum* extract caused the increase of intracellular space (4, 5 arrows) and the decrease of cytoplasm density observed as pale areas with lack of electron-dense (i.e., darker) cytoplasm (4, 5, asterisks). Lipid droplets contain fat with different electron density (5, L1 and L2). 7—SolA application caused a decrease in cytoplasm density (asterisk). 8—SolB application shows the decrease in electron density of lipid droplets. 9—SolC application caused a decline of cytoplasm electron density (asterisk).

**Figure 6 toxins-13-00617-f006:**
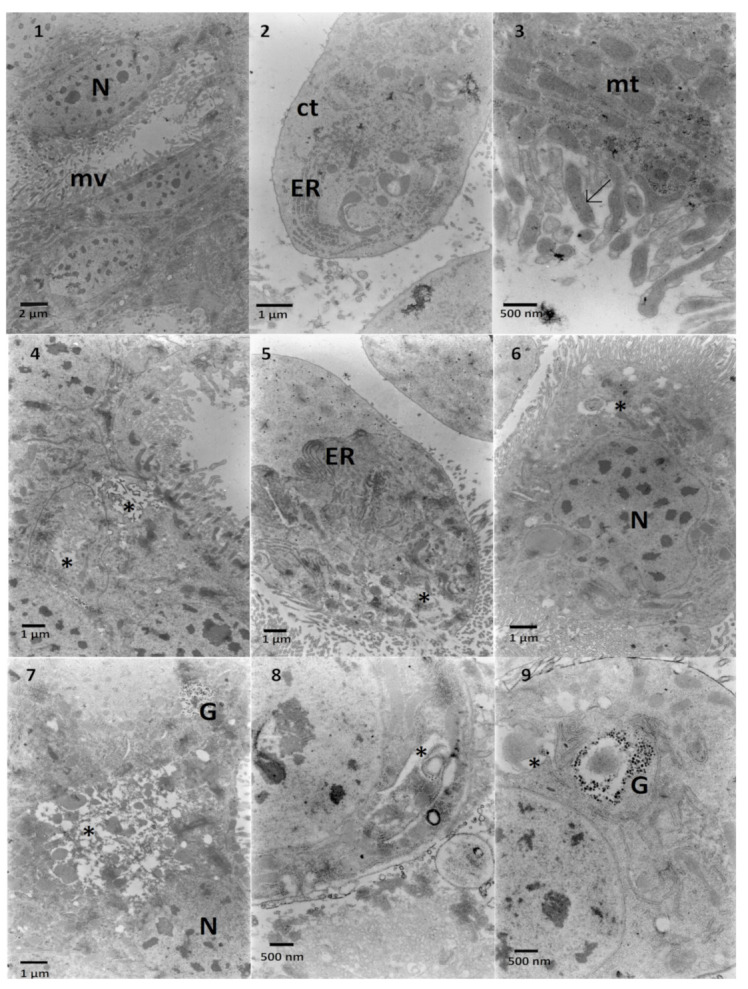
Midgut cells of *G. mellonella* larvae. 1–3—Control cells. Nucleus (N) and microvilli (mv) can be observed. Midgut cell (2) containing rich systems of endoplasmic reticulum (ER) in the cytoplasm (ct). Some mitochondria (mt) are located in microvilli (3, arrow). 4–7—*S. nigrum* extract given in the diet caused the appearance of vacuolized areas in the cytoplasm (4–7 asterisks) and increased amounts of the reticulum systems (5, ER). 8,9—SolA present in the larvae nourishment caused swelling in the endoplasmic reticulum (8, 9 asterisks). Glycogen granules can be observed (9, G).

**Figure 7 toxins-13-00617-f007:**
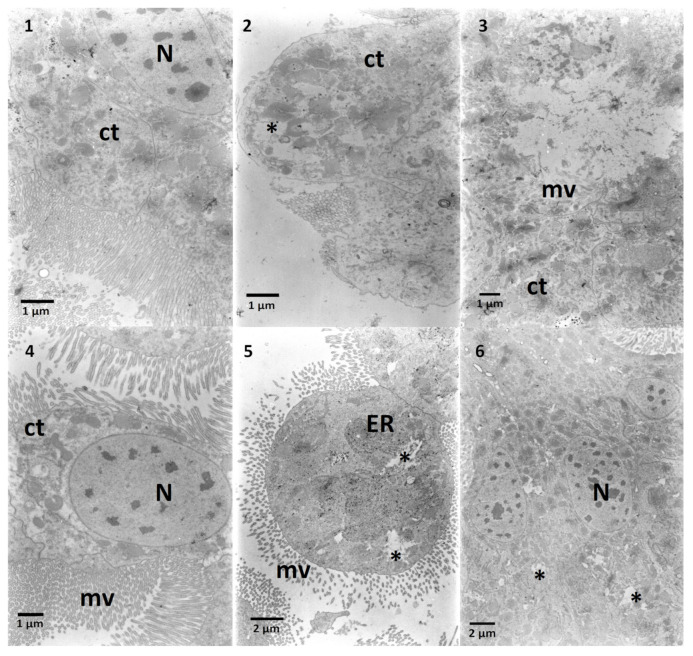
Ultrastructure of the midgut cells of *G. mellonella* larvae. 1–3—SolB added to the nourishment caused the decrease of the electron density of cytoplasm (1, 3, ct) with areas of stronger vacuolization (2, asterisk). 4–6—SolC caused similar results as the above mentioned SolB. Cytoplasm electron density decreased compared to control (4, ct), and in selected areas created wider spaces (5, 6, asterisks).

**Table 1 toxins-13-00617-t001:** The effects of *S. nigrum* extract (Ext) and solasonine (Sol) in three concentrations: A—two times higher than in the extract, B—the same amount as in the extract, C—two times lower than in the extract on proline, polyols, and sugars levels in the hemolymph of *G. mellonella* larvae.

	Proline	Sorbitol	Mannitol	Erythritol	Inositol	Glucose	Trehalose
Ext	↓ *	-	↓	↑ *	-	↓	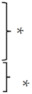	↓
SolA (2x↑Ext)	↓	-	↑	↑	↑	-	-
SolB (-II- Ext)	↓ *	-	↓	↑	-	↑	-
SolC (2x↓Ext)	↓	↓	↑	↑	-	↓	-

* An arrow marked with an asterisk shows statistically significant differences, without asterisk shows slight changes, and “-” indicates no changes. The buckle connects differences with statistical significance between particular concentrations.

## Data Availability

Not applicable.

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
