# Peer review of "Solanum nigrum Extract and Solasonine Affected Hemolymph Metabolites and Ultrastructure of the Fat Body and the Midgut in Galleria mellonella"

_toxins, 2021, doi:10.3390/toxins13090617_

Round 1

Reviewer 1 Report

Special comments

In what way represent this feeding experiment models the usual feeding habits of the beeswax moth? Do these animals come into contact with members of the Solanaceae family? I guess the answer is “no”, but it does not influence the merits of the study.

What I am more concerned about is that suppose the final end-product would be a natural, eco-friendly insecticide that kills the pest? Let’s suppose it is a bait drenched in the “harmful” solution, and those baits are placed within the hive. Why would the pest eat the bee products treated with the substance when there are non-treated bee products are also available within the same hive?

Is this substance harmful to bees? Or humans?

How can a bee-keeper separate which bee products are safe for sale and which are not?

Title

The Title reflects the content, relevant and is straightforward.

Abstract

The Abstract is within the word count limit.

However, the different sections of the Manuscript are represented disproportionately.

“Introduction” was given only one sentence, and that does not emphasize the importance and/or the main objective of the study. 

The “M+M” section is not represented at all. It is missing.

Results are over-represented. The reader does not want this much detail in an Abstract. 

Section “Conclusions” was given one sentence, too, and some further suggestions are lacking.

Keywords

The number of keywords is within the limit, and are specific to the manuscript, but I would recommend including words to refer to the natural insecticidal properties of botanicals. To attract readers coming from the field of crop protection. The Authors may choose from the following suggestions or create their own ones: “IPM”, “botanicals”, “organic insecticide”, “botanical insecticide”, “botanical-based crop protection”.

When these new keywords are added, some “old ones” need to be removed.

Key Contribution

It is a valid contribution. The Authors indeed selected the most important part of their manuscript.

Introduction

This Introduction is a bit short, and fairly comprehensible to researchers working outside the topic of the Manuscript. Unfortunately, the reason to investigate fat body and midgut remains unclear. I understand that these organs are responsible for digestion and detoxification. This makes them important objects of investigation, but the Authors should have elaborated on their role in in pest control (or in the use of natural insecticides). So this is where the Introduction needs a bit more additional information, some background. Also, the reader does not know why test sugars, polyols and proline levels? What’s the point?

It contains information that should have been positioned in the M+M section (Lines 37, and from Line 56 on.)

I suggest the Authors work on the flow of the text with the help of native speaker who is familiar with academic writing.

The Authors mention the objective of the study, although I would have welcomed more details about how this study would help to find solution to the problem.

Results

Changes to the haemolymph content (HPLC results) are clearly presented. The diagrams are informative, and the tables, especially Table 1. are useful.

The presentation of the effects on the ultrastructure needs some improvement.

Although there is explanation in the text, it is not always obvious on Figure 5, what has changed due to the treatment. When intracellular space increases for example, readers can’t compare what they see is abnormal, because the “normal” intracellular space is not shown. Also, it takes a lot of fantasy to see how the treatment caused a decline of cytoplasm electron density in Photo 9 of Figure 5. Not all readers are fluid interpreters of electronograms.

Discussion

Finally, the Discussion sheds some light on the questions I had during the Introduction and the Results. I still think some of these explanatory information should have been placed in the Introduction.

Line 205 mentions “Ext” – but no explanation is given as to its meaning.

The real Discussion starts at Line 240. I think the Authors should re-arrange the Discussion, probably by 1) removing some passages to the Introduction and by 2) introducing more material on the pest control aspect of the study. And/or on the ecologically friendly bee-keeping aspects, too.

Conclusions

Line 270: strange composition. It wasn’t the larvae that showed sublethal effects, but the treatment had sublethal effect on the larvae.

These conclusions are valid and are based on the experimental results.

M+M

Most of the description is correct and is easy to follow for those who are familiar with these methods.

How long the feeding last? (Days or hours?)

In what way did it model the usual feeding habits of the pest? Regarding time and usual amount eaten by the pest?

What was the reason behind creating this particular feeding material? Why was this particular concentration of the tested extract and solasonine used in the experiment (Line 239 0.5 mg per 1 g)?

What were the conditions of feeding? How did they model the conditions the larvae find within the beehive? I think the Authors should point out those conditions (temperature, light/darkness, humidity). What was the diet the larvae were on before starting the experiment?

Line 309 informs the reader that larvae fed with solvent is the control. What exactly this solvent is?

Yet in Line 316, there are three groups of larvae: one fed with the extract, one with pure glycoalkaloids, and there is the control. How come there are three feeding groups now? Please be more specific when describing the treatment.

Line 316 “Larvae fed with nourishment containing the extract” this composition is quite awkward. Suggestion: “Larvae fed with the extract” OR “treated larvae”.

References

Literature is formatted according to the rules, except for No. 25.

The number of references is a bit low, only 38.

This Manuscript uses old and present-day literature as well. 28 of the papers were published after 2000.

The number of self-references can not be evaluated, as the Authors are unknown.

Author Response

In what way represent this feeding experiment models the usual feeding habits of the beeswax moth? Do these animals come into contact with members of the Solanaceae family? I guess the answer is “no”, but it does not influence the merits of the study. What I am more concerned about is that suppose the final end-product would be a natural, eco-friendly insecticide that kills the pest? Let’s suppose it is a bait drenched in the “harmful” solution, and those baits are placed within the hive. Why would the pest eat the bee products treated with the substance when there are non-treated bee products are also available within the same hive? Is this substance harmful to bees? Or humans? How can a bee-keeper separate which bee products are safe for sale and which are not?

The study was performed on G. mellonella which in this case was used as the model species for the whole Lepidoptera order which involves a huge number of pests. Thus, we do not address the solution about how, in the human safe and efficient way, to expose the G. mellonella to the tested substances in the hives. The main aim was to check what will be the effect of glycoalkaloids on chosen tissues of the insect. The concerns of the reviewer are thus understandable and probably need to be addressed in a different study, as another step – selectivity of the tested substance and mode of its application so that it would be harmful to the pests but not the bees. Indeed has been reported that alkaloids of the Nicotiana glauca Graham (Solanaceae) have low levels of toxicity on bees, causing the 16–18% of mortality. However, we added some information in the introduction to clarify the usage of G. mellonella as a model.

The Abstract is within the word count limit.

However, the different sections of the Manuscript are represented disproportionately.

“Introduction” was given only one sentence, and that does not emphasize the importance and/or the main objective of the study.

The “M+M” section is not represented at all. It is missing.

Results are over-represented. The reader does not want this much detail in an Abstract.

Section “Conclusions” was given one sentence, too, and some further suggestions are lacking.

We thank the reviewer for the observation, we changed the abstract accordingly.

The number of keywords is within the limit, and are specific to the manuscript, but I would recommend including words to refer to the natural insecticidal properties of botanicals. To attract readers coming from the field of crop protection. The Authors may choose from the following suggestions or create their own ones: “IPM”, “botanicals”, “organic insecticide”, “botanical insecticide”, “botanical-based crop protection”.

When these new keywords are added, some “old ones” need to be removed.

The keyword “botanical insecticide” has been added, and the keyword “sugars” was removed.

This Introduction is a bit short, and fairly comprehensible to researchers working outside the topic of the Manuscript. Unfortunately, the reason to investigate fat body and midgut remains unclear. I understand that these organs are responsible for digestion and detoxification. This makes them important objects of investigation, but the Authors should have elaborated on their role in in pest control (or in the use of natural insecticides). So this is where the Introduction needs a bit more additional information, some background. Also, the reader does not know why test sugars, polyols and proline levels? What’s the point?

It contains information that should have been positioned in the M+M section (Lines 37, and from Line 56 on.)

I suggest the Authors work on the flow of the text with the help of native speaker who is familiar with academic writing.

The Authors mention the objective of the study, although I would have welcomed more details about how this study would help to find solution to the problem.

Thank you for this comment, this will surely make the manuscript more clear. The introduction has been extended with necessary information and lines from 56 on were moved to M&M.

Changes to the haemolymph content (HPLC results) are clearly presented. The diagrams are informative, and the tables, especially Table 1. are useful.

The presentation of the effects on the ultrastructure needs some improvement.

Although there is explanation in the text, it is not always obvious on Figure 5, what has changed due to the treatment. When intracellular space increases for example, readers can’t compare what they see is abnormal, because the “normal” intracellular space is not shown.

On the Figure 5, No 2. the control cells adhere evenly and it is marked with an arrow, and it is described in the text. Therefore, it can be compared with the changes presented on the other electronograms after glycoalkaloids treatment. We are aware that particular changes could be shown better, however we wanted to avoid putting too many electronograms and to show few features on one, instead.

Also, it takes a lot of fantasy to see how the treatment caused a decline of cytoplasm electron density in Photo 9 of Figure 5. Not all readers are fluid interpreters of electronograms.

We agree that transmission electron microscopy is a difficult technique and it takes years to learn how to interpret observed changes. On the electronograms we tried to show and point as many observations as possible. Comparing the density of cytoplasm was not based on any calculation, but on the eye of experienced scientists. For example, on the Figure 5, No 6., two different cells are shown; one with electron dense cytoplasm, and the other with electron lucent cytoplasm (upper, with an asterisk). On the Figure 5 No. 9 part of the cytoplasm is marked with an asterisk, which shows the particular areas with significant decrease of the cytoplasm electron-density.

Finally, the Discussion sheds some light on the questions I had during the Introduction and the Results. I still think some of these explanatory information should have been placed in the Introduction.

Line 205 mentions “Ext” – but no explanation is given as to its meaning.

We thank the reviewer for his/her comment. “Ext” is the abbreviation of Extract, as we reported the first time in the result section “2.1.2. Polyols”. We explained it also in the discussion section.

The real Discussion starts at Line 240. I think the Authors should re-arrange the Discussion, probably by 1) removing some passages to the Introduction and by 2) introducing more material on the pest control aspect of the study. And/or on the ecologically friendly bee-keeping aspects, too.

Thank you for this observation. The Introduction and Discussion were corrected. In the introduction we added missing information about the pest control aspects, also about the reason why we chose the fat body and the midgut in our experiments. In the discussion we added two literature positions about the effects of glycoalkaloids on bees and the possibility to use Solanaceae as pest control. We think that the Discussion, where we compared the results with other researchers, is the best place to describe a potential reason of our observation. We think that moving the first part of the Discussion into the Introduction would make the Introduction too extensive therefore we didn’t move it. We hope that all the other improvements of our manuscript made it clearer and more readable.

Line 270: strange composition. It wasn’t the larvae that showed sublethal effects, but the treatment had sublethal effect on the larvae.

These conclusions are valid and are based on the experimental results.

We thank the reviewer for the observation, we changed accordingly.

Most of the description is correct and is easy to follow for those who are familiar with these methods.

How long the feeding last? (Days or hours?)

The feeding takes 3 days. This information is already included in the M&M section.

In what way did it model the usual feeding habits of the pest? Regarding time and usual amount eaten by the pest?

The experiment was conducted in closed plastic flasks placed in the same conditions as they were kept before the experiment. We didn’t calculate the amount of the diet eaten by each larva. Since in the plastic tube we placed 6 larvae the amount of eaten food is an individual feature. However some of the diet remains not eaten after the end of experiment.

What was the reason behind creating this particular feeding material? Why was this particular concentration of the tested extract and solasonine used in the experiment (Line 239 0.5 mg per 1 g)?

The feeding material imitates the natural habitat of the pest. The S. nigrum extract wasn’t tested on G. mellonella before, but glycoalkaloid α-solanine was tested (Büyükgüzel et al. The influence of dietary α-solanine on the wax moth Galleria Mellonella L. Arch Insect Biochem Physiol 2013, 83(1):15-24.). We have chosen a concentration of solasonine similar to sublethal concentrations of solanine used in the experiment of Büyükgüzel et al. We created proper extract concentration on the basis of how much of solasonine was present as the extract’s ingredient.

What were the conditions of feeding? How did they model the conditions the larvae find within the beehive? I think the Authors should point out those conditions (temperature, light/darkness, humidity). What was the diet the larvae were on before starting the experiment?

These data are provided in the M&M section.

Line 309 informs the reader that larvae fed with solvent is the control. What exactly this solvent is?

The solvent was 1% formic acid as the solvent for glycoalkaloids. We added this information.

Yet in Line 316, there are three groups of larvae: one fed with the extract, one with pure glycoalkaloids, and there is the control. How come there are three feeding groups now? Please be more specific when describing the treatment.

The reviewer is right, the sentence is not clear. There are three main groups: larvae fed with diet containing the extract, larvae fed with diet containing the pure glycoalkaloids at different concentrations and larvae control. We provided the information also in this part of the text.

Line 316 “Larvae fed with nourishment containing the extract” this composition is quite awkward. Suggestion: “Larvae fed with the extract” OR “treated larvae”.

It was corrected.

Literature is formatted according to the rules, except for No. 25.

We changed the reference according to the author’s guidelines.

The number of references is a bit low, only 38.

Two new references have been added, numbers 37 and 38.

This Manuscript uses old and present-day literature as well. 28 of the papers were published after 2000.

The number of self-references can not be evaluated, as the Authors are unknown.

We would like to thank the Reviewer for the valuable remarks.

Reviewer 2 Report

This manuscript is of good scientific quality, however, the authors need to improve it. The manuscript needs grammar editing and the materials and methods part of this paper needs more clarity.

Lines 71, 75, 76: Grammar check.

Line 216: Grammar check

Line 294: The extract from S. nigrum berries was prepared obtained from prof. Sabino A. Bufo, Potenza, Italy.

Was the extract prepared by the authors or obtained from another source?

Lines 308-309: "Or" is a conjunction used to imply the possibility of one thing. The use of "or" in the following sentence: "Six IVth instar larvae were placed in a plastic tube containing 10 g of the diet mixed with tested substances or with a solvent as a control" suggests the authors are choosing between the experiment and the control which is not the case here.

The phrase "placed in a plastic tube" also suggests that the six IVth instar larvae were all placed in one plastic tube. If this is not the case, kindly clarify.

Line 337: Grammar check

Author Response

This manuscript is of good scientific quality, however, the authors need to improve it. The manuscript needs grammar editing and the materials and methods part of this paper needs more clarity.

Lines 71, 75, 76, 216, 337: Grammar check

We rephrased the sentences.

Line 294: The extract from S. nigrum berries was prepared obtained from prof. Sabino A. Bufo, Potenza, Italy.

Was the extract prepared by the authors or obtained from another source?

The extract was prepared by the authors in the laboratory of the co-authors Prof. S.A. Bufo. We corrected in the manuscript.

Lines 308-309: "Or" is a conjunction used to imply the possibility of one thing. The use of "or" in the following sentence: "Six IVth instar larvae were placed in a plastic tube containing 10 g of the diet mixed with tested substances or with a solvent as a control" suggests the authors are choosing between the experiment and the control which is not the case here.

The sentence was changed.

The phrase "placed in a plastic tube" also suggests that the six IVth instar larvae were all placed in one plastic tube. If this is not the case, kindly clarify.

Yes, correctly, 6 larvae were placed in one tube, we clarified it.

We would like to thank to the Reviewer for the valuable remarks.

Reviewer 3 Report

The manuscript submitted for review contains interesting research findings on the influence of a crude extract obtained from Solanum nigrum and solasonine on the level of hemolymph metabolites such as sugars, polyols and proline, and ultrastructure of the fat body and the midgut cells. G. mellonella larvae were used as an insect model. The research methods of the experiments were well planned and the results clearly presented. The information on the statistical methods used in particular requires elaboration. It is also important to develop the hypothesis in the discussion that compounds from Solanum nigrum can be used as insecticides. Detailed comments are provided below.

Lines 41-44 Do you know the effect of Solanaceae plants  metabolites on bees?

Line 88, 105, 107, 119 Please give p-value and n-value for statistically significant differences

Fig. 2, 3, 4. Do I understand correctly that the data is presented as mean and standard deviation? Please provide the relevant information in the legend.

Lines 297 and 302-306 Why was this concentration of extract and solasonine chosen?

Why was the two times higher/lower than the extract concentration, for example, not three or ten times?

Line 318 You wrote that the hemolymph was taken from 5 larvae. The previous paragraph (line 308) states that 6 larvae were used for each experiment. Please, clarify.

Section “Materials and methods”. No information on the statistical methods used. Please provide information on the tests used to test the normality of the distribution and the statistical significance of the differences. What software was used for the statistical analysis?

In conclusion, the authors emphasized the possibility of potential usage of glycoalkaloids as bioinsecticides. Did the authors determine the effect of the studied extract and solasonine on larvae mortality? From the word "sublethal" used in the conclusion, I suppose that the mortality rate was very low. Therefore I ask for clarification, which may be sublethal effects of the use of selected substances to insect (based on your observation or literature)? The exposure of the larvae to food with the addition of an extract or solasonine lasts for 3 days, therefore the effect on pupation or imaginary moult could not be determined. Therefore, are there data in the literature on the influence of compounds from similar chemical groups on the metamorphosis process in G. mellonella or other insects?

Author Response

The manuscript submitted for review contains interesting research findings on the influence of a crude extract obtained from Solanum nigrum and solasonine on the level of hemolymph metabolites such as sugars, polyols and proline, and ultrastructure of the fat body and the midgut cells. G. mellonella larvae were used as an insect model. The research methods of the experiments were well planned and the results clearly presented. The information on the statistical methods used in particular requires elaboration. It is also important to develop the hypothesis in the discussion that compounds from Solanum nigrum can be used as insecticides. Detailed comments are provided below.

Lines 41-44 Do you know the effect of Solanaceae plants metabolites on bees?

Just few studies are reported on the effects of Solanaceae plant metabolites on bees. We reported these studies in the discussion section.

Line 88, 105, 107, 119 Please give p-value and n-value for statistically significant differences

The values were added.

Fig. 2, 3, 4. Do I understand correctly that the data is presented as mean and standard deviation? Please provide the relevant information in the legend.

We added the missing information. The data on graphs present mean value ± SEM.

Lines 297 and 302-306 Why was this concentration of extract and solasonine chosen?

The S. nigrum extract wasn’t tested on G. mellonella before, but glycoalkaloid α-solanine was tested (Büyükgüzel et al. The influence of dietary α-solanine on the wax moth Galleria Mellonella L. Arch Insect Biochem Physiol 2013, 83(1):15-24.). We have chosen a concentration of solasonine similar to sublethal concentrations of solanine used in the experiment of Büyükgüzel et al. We created proper extract concentration on the basis of how much of solasonine we obtained as its ingredient.

Why was the two times higher/lower than the extract concentration, for example, not three or ten times?

The used concentration was chosen to check the potential of glycoalkaloid mode of action as an extract ingredient. We were interested in the low concentrations because of future economic reasons. However the use of higher concentrations is under consideration in future studies.

Line 318 You wrote that the hemolymph was taken from 5 larvae. The previous paragraph (line 308) states that 6 larvae were used for each experiment. Please, clarify.

Yes it is correct, one larvae was randomly rejected. The final concentration of hemolymph in one the sample was then 10 µL.

Section “Materials and methods”. No information on the statistical methods used. Please provide information on the tests used to test the normality of the distribution and the statistical significance of the differences. What software was used for the statistical analysis?

We added the section 5.8 Statistical analysis.

In conclusion, the authors emphasized the possibility of potential usage of glycoalkaloids as bioinsecticides. Did the authors determine the effect of the studied extract and solasonine on larvae mortality?

We tested only the concentrations given in the manuscript which were sublethal. However we think the LC values could be checked and, as mentioned before by the Reviewer, higher concentrations can be tested in the future studies.

From the word "sublethal" used in the conclusion, I suppose that the mortality rate was very low.

There were no lethal effects observed during the experiment.

Therefore I ask for clarification, which may be sublethal effects of the use of selected substances to insect (based on your observation or literature)? The exposure of the larvae to food with the addition of an extract or solasonine lasts for 3 days, therefore the effect on pupation or imaginary moult could not be determined. Therefore, are there data in the literature on the influence of compounds from similar chemical groups on the metamorphosis process in G. mellonella or other insects?

Thank you for this question. The short exposure on the tested extract was checked in our studies previously, although on a beetle, but not only in the context of using the extract or glycoalkaloids as single bioinsecticides, but also as a factor decreasing the ability of insects to detoxify synthetic insecticides. Our previous results have shown that the extract can weaken the organism of an insect and make it more vulnerable to applied insecticides.

The exposure of pupae on the injected glycoalkaloids caused the disturbances of myocardium contractions (Marciniak, P.; KoliÅ„ska, A.; Spochacz, M.; ChowaÅ„ski, S.; Adamski, Z.; Scrano, L.; Falabella, P.; Bufo, S.A.; RosiÅ„ski, G. Differentiated Effects of Secondary Metabolites from Solanaceae and Brassicaceae Plant Families on the Heartbeat of Tenebrio molitor Pupae. Toxins 2019, 11, 287), which can be an indication of further disturbations in normal development. Weissenberg et al. (Weissenberg, M.; Levy, A.; Svoboda, J.A.; Ishaaya, I. The effect of some solanum steroidal alkaloids and glycoalkaloids on larvae of the red flour beetle, Tribolium castaneum, and the tobacco hornworm, Manduca sexta. Phytochemistry 1998, 47, 203–209) showed in their studies that glycoalkaloid tomatine inhibited the development of the other Lepidoptera, Manduca sexta. Therefore we suspect the extract we used in the studies may have a similar impact on G. mellonella.

We would like to thank to the Reviewer for the valuable remarks.